# Structural basis of soluble membrane attack complex packaging for clearance

Anaïs Menny [1,4], Marie V. Lukassen[2,3,4], Emma C. Couves[1], Vojtech Franc[2,3], Albert J. R. Heck [2,3] & Doryen Bubeck [1✉]

Unregulated complement activation causes inflammatory and immunological pathologies with consequences for human disease. To prevent bystander damage during an immune response, extracellular chaperones (clusterin and vitronectin) capture and clear soluble precursors to the membrane attack complex (sMAC). However, how these chaperones block further polymerization of MAC and prevent the complex from binding target membranes remains unclear. Here, we address that question by combining cryo electron microscopy (cryoEM) and cross-linking mass spectrometry (XL-MS) to solve the structure of sMAC. Together our data reveal how clusterin recognizes and inhibits polymerizing complement proteins by binding a negatively charged surface of sMAC. Furthermore, we show that the pore-forming C9 protein is trapped in an intermediate conformation whereby only one of its two transmembrane β-hairpins has unfurled. This structure provides molecular details for immune pore formation and helps explain a complement control mechanism that has potential implications for how cell clearance pathways mediate immune homeostasis.

[1] Department of Life Sciences, Sir Ernst Chain Building, Imperial College London, London SW7 2AZ, UK. [2] Biomolecular Mass Spectrometry and Proteomics, Bijvoet Center for Biomolecular Research and Utrecht Institute for Pharmaceutical Sciences, Utrecht University, Padulaan 8, 3584 CH Utrecht, The Netherlands. [3] Netherlands Proteomics Center, Padulaan 8, 3584 CH Utrecht, The Netherlands. [4]These authors contributed equally: Anaïs Menny, Marie V. Lukassen. ✉email: d.bubeck@imperial.ac.uk

The complement membrane attack complex (MAC) is an immune pore that directly kills pathogens and causes human disease if left unchecked. One of the first lines of defense against Gram-negative bacteria[1], MAC is a potent weapon of the innate immune system that can rupture lipid bilayers of any composition. Therefore, MAC is highly regulated on human cells to prevent damage[2,3]. An inhibitory protein blocks MAC assembly and pore formation that occurs directly on the plasma membrane of human cells[4]. However, in the absence of inhibitory plasma proteins, complement complexes that have not assembled on target membranes can lyse red blood cells[5]. When released from complement-opsonized pathogens, these complexes can also deposit on nearby macrophages and initiate a cascade of inflammatory responses causing bystander damage[6]. Therefore, understanding how MAC is controlled is essential for our ability to tune the activity of a potent innate immune effector and prevent human disease.

Soluble MAC (sMAC) is an immune activation complex that is formed from MAC assembly precursors released into plasma and scavenged by blood-based chaperones. While in healthy individuals sMAC exists in trace amounts, these levels are dramatically elevated during an immune response, providing a biomarker for infectious and autoimmune disease[7,8], transplant[9,10], and trauma[11]. sMAC, also known as sC5b9, is composed of the complement proteins C5b, C6, C7, C8, and C9 together with the extracellular regulatory proteins, clusterin, and vitronectin[12,13]. Derived from MAC precursors, sMAC is a model system for understanding structural transitions underpinning MAC assembly. In both sMAC and MAC, complement proteins associate through their pore-forming membrane attack complex perforin (MACPF) domain[14,15]. Structures of MAC[15,16] and soluble C9[17] show that complement proteins undergo substantial conformational rearrangements to enable pore formation. While these studies have contributed to our understanding of the final pore, little is known of how regulators trap transition states and clear activation byproducts.

Clusterin (also called apolipoprotein J) is a chaperone that broadly protects against pathogenic aggregation of proteins. Upregulated in response to cellular stress[18], clusterin recognizes a variety of cellular targets and traffics cargo for disposal. Within sMAC, clusterin binds fluid-phase oligomeric complement complexes generated during an immune response[12] and inhibits polymerization of C9[19]. The chaperone also directly associates with amyloid-beta fibrils formed from the polymerizing Aβ peptide and prevents peptide aggregation[20,21]. Indeed, mutation of the gene that encodes human clusterin, *CLU*, is one of the greatest genetic risk factors for late-onset Alzheimer's disease[22]. Although the roles of clusterin in protein quality control and clearance pathways are well established, the molecular mechanism by which clusterin recognizes and traffics cargo for degradation remains unclear.

To understand how fluid-phase chaperones trap and clear MAC assembly intermediates, we combined cryo electron microscopy (cryoEM) and cross-linking mass spectrometry (XL-MS) to solve the structure of sMAC. We find that sMAC is a heterogeneous complex in which the pore-forming MACPF domain of C9 is caught in a transition state. We also provide a molecular basis for how C7 may activate C5b to propagate MAC assembly. Finally, we discover that clusterin can bridge MAC proteins through electrostatic interactions and obstruct the polymerizing face of C9. Taken together, these data provide a structural framework for understanding pore formation and the molecular details for a complement control mechanism.

## Results

**sMAC is a heterogeneous multi-protein complex.** Complement activation produces soluble protein assemblies that are cleared by chaperones in blood plasma. To understand how these immune activation macromolecules are trapped and shuttled for removal, we used cryoEM to visualize sMAC (Fig. 1 and Supplementary Fig. 1). In accordance with lower resolution studies[14], we find that C5b supports the assembly of complement proteins (C6, C7, C8, and C9) into an arc-like arrangement of MACPF domains. We discover that sMAC extends 260 Å in length with clear density for extended β-hairpins of MACPF domains. Similar to MAC[15], sMAC contains a single copy of the complement proteins C5b, C6, C7, and C8 while the stoichiometry of C9 varied between one to three copies (Supplementary Fig. 1). Using 3D classification of cryoEM images, we generated reconstructions for complexes with either one, two or three copies of C9 to a resolution of 3.8 Å, 3.3 Å, and 3.5 Å, respectively (Figs. 1a, b and Supplementary Fig. 1). Anisotropy in 1C9-sMAC reconstruction prevented further modeling of this complex; however, cryoEM density maps for the 2- and 3-C9 sMAC complexes were sufficient to build a near-complete atomic model for all complement proteins (Figs. 1a, b, Supplementary Fig. 2 and Supplementary Table 1). Density for clusterin and vitronectin were not well resolved in these initial maps, consistent with the flexible nature of these chaperones.

To confirm the composition and structure of proteins in sMAC, we performed label-free quantitative proteomics and XL-MS on sMAC. Our proteomics analysis confirmed the presence of all complement proteins (C5b, C6, C7, C8, and C9), with C9 twice as abundant as the others (Supplementary Fig. 3). In agreement with sedimentation centrifugation measurements[13], our analyses also revealed the presence of clusterin and vitronectin with abundances 2-6 times higher than that of C5b, C6, C7, and C8 (Supplementary Fig. 3). All other proteins detected in the sMAC sample were an order of magnitude less abundant. In the XL-MS data we observed a number of intra- and inter-protein cross-links between complement proteins, with 88% fitting the distance restraints of the atomic models derived from our sMAC reconstructions (Supplementary Fig. 4 and Source data file). Furthermore, we observed several inter-links between complement proteins, vitronectin and clusterin, confirming their presence in the complex (Fig. 1c, Supplementary Fig. 3, and Source data file). Mass photometry of sMAC showed a handful of co-occurring multi-protein complexes with distinct masses between 971 and 1374 kDa (Fig. 1d). After subtracting masses corresponding to the core complement complex (C5b6, C7, C8) together with one, two or three C9 molecules, we are left with excess masses of minimally ~241 and maximally ~777 kDa. As the molecular weights of secreted clusterin and vitronectin are ~80 kDa and ~75 kDa, respectively, we attribute this excess to multiple copies of both these chaperones. These data confirm that sMAC is a heterogeneous assembly comprising a single copy of a conserved core complex (C5b6, C7, and C8) together with multiple copies of C9, clusterin and vitronectin in a mixture of stoichiometries.

**Clusterin bridges complement proteins through electrostatic interactions.** With the high abundance of clusterin and vitronectin in sMAC, we next sought to identify their location in the complex. Given the extensive cross-links observed between the chaperones and C9 (Fig. 1c, Supplementary Fig. 3, and Source data file), we subtracted density corresponding to proteins C5b6, C7, and C8 from the raw EM images and focused our refinement on the C9 component of the complex (Fig. 2a). In doing so, we resolve a 10 nm stretch of density that bridges the negatively charged crown of C9 lipoprotein receptor class A (LDL) domains. This density caps a similarly negatively charged polymerizing face of the C9 MACPF (Fig. 2c, d) and sterically obstructs additional

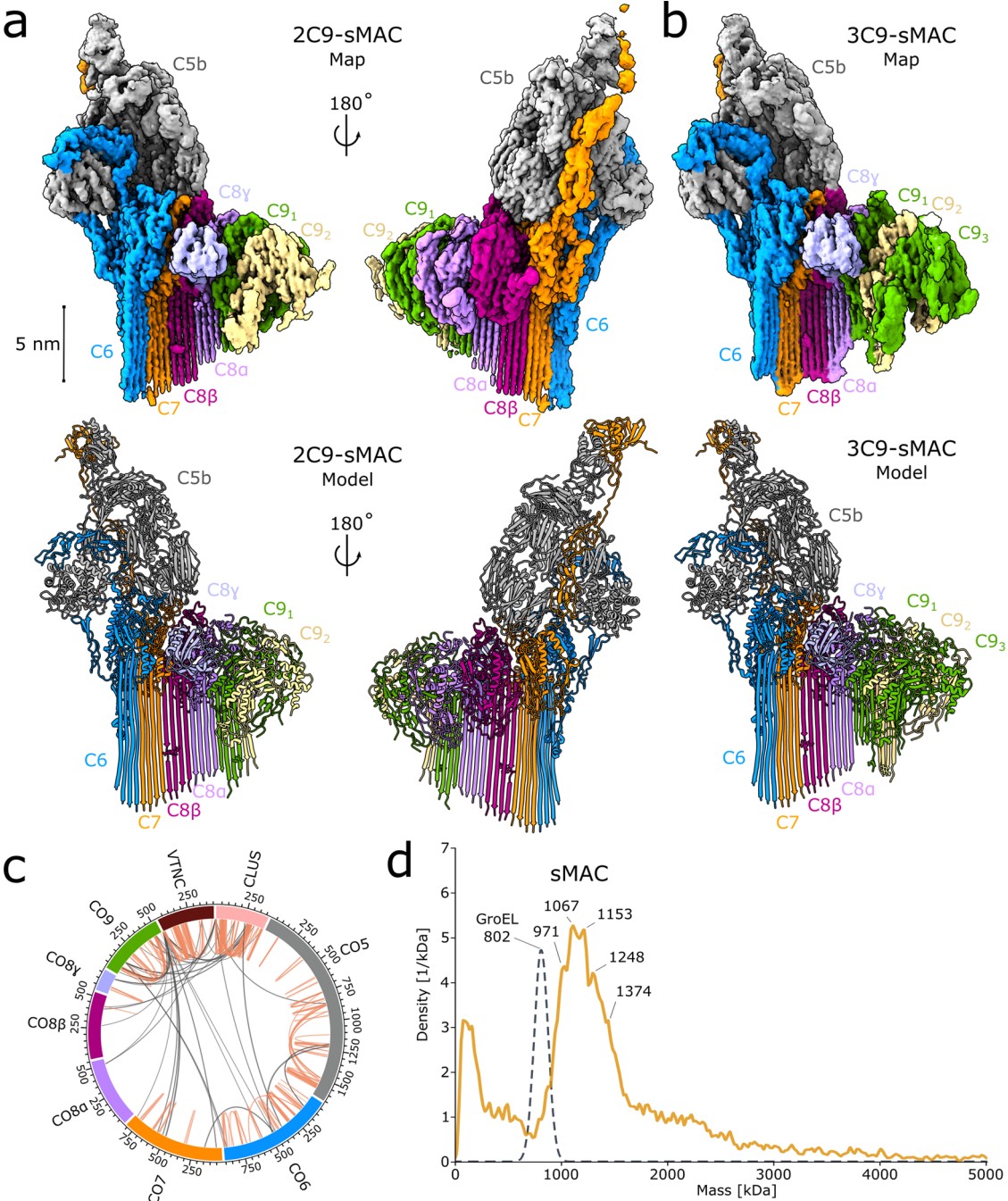

**Fig. 1 sMAC is a complement activation macromolecule with a heterogeneous composition. a** CryoEM reconstruction (top) and atomic model (bottom) of sMAC that consists of a core complement complex (C5b, C6, C7, and C8α/C8β/C8γ) together with two C9 molecules (C9₁, C9₂). **b** CryoEM reconstruction (top) and atomic model (bottom) of sMAC with the same core complement complex and three molecules of C9 (C9₁, C9₂, C9₃). CryoEM density maps in a and b are colored according to protein composition. Glycans included in the atomic models are shown as sticks in the ribbon diagrams. **c** sMAC circos plot of identified 4-(4,6-Dimethoxy-1,3,5-triazin-2-yl)-4-methylmorpholinium chloride (DMTMM) cross-links within and across complement components (C5, C6, C7, C8, and C9). The complement components are cross-linked to the chaperones vitronectin (VTNC) and clusterin (CLUS). Intra-links are shown as orange lines and inter-links are shown as gray lines. Source data are provided as a Source Data file. **d** Mass photometry of sMAC reveals a heterogeneous mixture of complexes (orange line). Masses of the most abundant complexes are indicated. Gaussian distribution of GroEL from the protein standard is shown as a reference for peak width of a homogeneous complex (gray dotted line).

C9 binding (Supplementary Fig. 5). We also observe a second belt of density below the C9 epidermal growth factor (EGF) domains (Fig. 2c). Locally sharpened maps indicate tubular densities consistent with alpha helices; however, this region remains highly flexible and we were unable to model it based on our cryoEM maps.

We next considered the possible identity of this extra density. All complement proteins within sMAC are accounted for within the map; therefore, we conclude that this density is composed of plasma-based chaperones. Based on our XL-MS data, both clusterin and vitronectin interact with C9 (Fig. 1c and Supplementary Fig. 3). We then plotted the location of

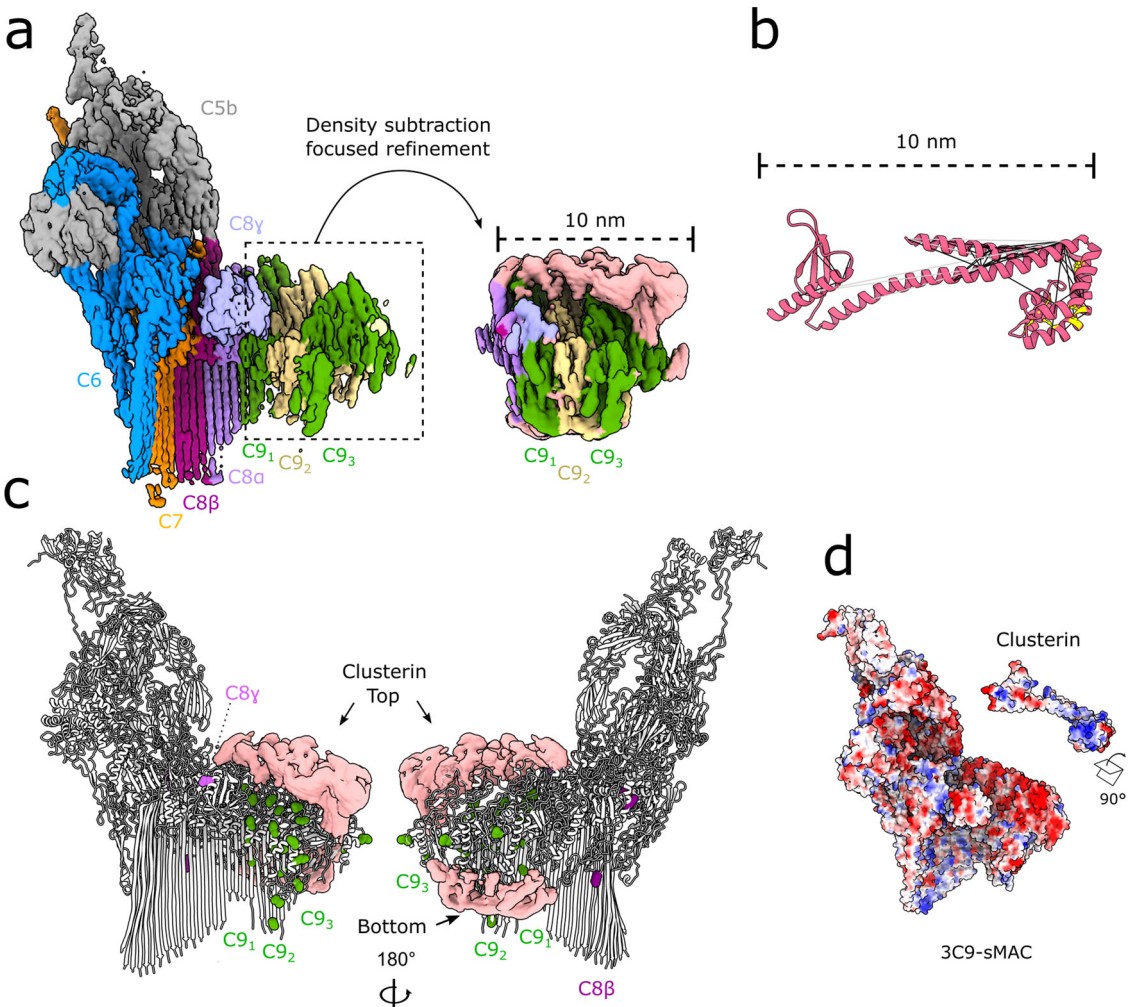

**Fig. 2 Clusterin bridges complement proteins in sMAC through electrostatic interactions. a** CryoEM reconstruction of 3C9-sMAC highlighting the region used in subsequent focused refinements (black-dotted lines). Inset shows the map after density subtraction of the core complement complex and refinement of the C9 oligomer. Density is colored according to protein composition, with regions of the map not accounted for by complement proteins in pink. **b** Structural homology model for the clusterin core (pink ribbons) with intra-molecular clusterin cross-links derived from XL-MS mapped (black lines). Over-length cross-links are shown as gray lines and known disulfide bonds within clusterin are shown in yellow. Source data are provided as a Source Data file. **c** Intermolecular cross-links between clusterin and C8/C9 plotted on the 3C9-sMAC model (white ribbons). Complement protein residues involved in cross-links are shown as spheres colored according to protein composition. Density not corresponding to complement proteins in the focused refined map (pink surface) is overlaid for reference. **d** Coulombic electrostatic potential ranging from −10 (red) to 10 (blue) kcal/(mol·e) calculated from the models for complement proteins in 3C9-sMAC (bottom surface) and the model for clusterin shown in 2b (top surface).

complement protein residues in our 3C9-sMAC model that uniquely cross-link to each chaperone (Supplementary Fig. 3). We find that cross-links between clusterin and C9 congregate around the two extra densities observed in our focus-refined map (Fig. 2c). By contrast, unique cross-links between vitronectin and C9 lie on the back face of the map (Supplementary Fig. 3). These residues are not resolved in the final C9 of the 3C9-sMAC reconstruction; therefore, it is also possible that they reflect cross-links to the mobile terminal C9 in other sMAC stoichiometries. Additionally, we find that the number of cross-links between vitronectin and C9 are identified about 3.5 times less than those observed to clusterin. As the cross-linkers used in these experiments bridge charged residues (Lys-Lys or Asp/Glu-Lys), these data could indicate that vitronectin binds to a hydrophobic region of C9, which is in agreement with immuno-gold labeling experiments that map the vitronectin bind-site to the MACPF hydrophobic hairpins[23]. While we observe some density for this region in other focused refinements (Supplementary Fig. 1), the area is not well ordered in our maps. Taken together, our data

demonstrate that clusterin likely occupies some or all of the extra density present above the C9 LDL domains in our focus-refined map.

Clusterin is a highly glycosylated[24], disulfide-linked heterodimer[25] whose tertiary structure remains unknown. Using trRosetta[26], which combines co-evolutionary data with deep learning, we generated a panel of possible clusterin structural models (Supplementary Fig. 6). All models contained a common helical core component with structural similarity to the MAC inhibitor CspA from *Borrelia burgdoferi*[27,28] (Supplementary Fig. 6). The clusterin core model extends 10 nm in length and is consistent with the density in our focus-refined map (Fig. 2a, b). To further optimize our model, we then applied known disulfide-bond restraints for clusterin using Modeller[29]. We next assessed the validity of our model by plotting the unique intra-molecular clusterin cross-links observed in our XL-MS experiments (Fig. 2b). In doing so, we verify that 92% of clusterin cross-links satisfy the distance restraints of our model (Supplementary Fig. 6 and Source Data file). Our analyses show that clusterin contains an

extended helical domain capped by a helical bundle, whose arrangement is defined by five disulfide bonds (Fig. 2b). The electrostatic surface potential of the clusterin core model reveals a contiguous patch of positive charge, complementary to the surface charge of C9 at the interface with clusterin in sMAC (Fig. 2d). Beyond this core, we find that clusterin makes additional cross-links with a range of complement proteins distal from C9 (Supplementary Fig. 3). Indeed, the ensemble of models produced by trRosetta reveals long extended domains that flexibly hinge from the core (Supplementary Fig. 6). It is possible that these extensions could interact with distal complement proteins. Altogether, our data are consistent with a model that clusterin is a highly flexible protein, whose core domain engages cargo through electrostatic interactions to block propagation of polymerizing proteins.

**C-terminal domains of C7 position the C345C domain of C5b.** Previous structural studies of MAC were limited in resolution due to the flexibility and varied curvature of the complex[16]. In particular, the C-terminal domains of C7 and their interaction with C5b remained unmodeled. As a result, it remained unclear how C7 activates C5b6 to propagate MAC assembly. To understand how the C7 C-terminal domains prime the complex for C8 recruitment, we sought to improve the resolution of the map in this area. Density for C5b was best resolved in our 2C9 sMAC map; therefore, we used this map for subsequent refinement steps in which density corresponding to the MACPF arc was subtracted (Fig. 3a). By focusing our alignment on C5b, we calculated a map with a resolution of 3.6 Å which enabled us to build a near complete atomic model for sMAC incorporated C7. By contrast to the extensive interaction interface between the C6 complement control protein (CCP) domains and C5b, we find that the binding site of C5b and the C-terminal CCPs of C7 is punctuated by three specific contact points (Fig. 3b). In our map, we observe clear side-chain density for ionic interactions between C7:$Asn_{572}$/$Arg_{590}$ and Cb5:$Gln_{73}$, which appears to stabilize the position of the first C7 CCP. We also observe unambiguous density for C5b:$Trp_{581}$ that wedges into a hydrophobic groove connecting the two CCPs and likely impacts the orientation of these domains. Stabilized by a glycan on C5b ($Asn_{893}$), a flexible linker connects the C7 CCPs and the final two factor I-like membrane attack complex (FIM) domains. Our data show that the first FIM domain is responsible for binding the C345C domain of C5b (Fig. 3a, d). Superposition of C5b from the soluble C5b6[14] and its conformation in sMAC shows that this domain undergoes the largest movement during MAC assembly (Fig. 3d). Interestingly, we find that the orientation of the C345C domain of C5b in our model overlays with its position in a structural homolog, C3b, when bound to Factor B and properdin in an activated conformation[30] (Fig. 3e). In addition, we find that the Macroglobulin (MG) 4 and MG5 domains of C5b also move to accommodate C8 allowing a network of salt bridges between the loops of the C5b MG scaffold and the LDL domain of C8β (Fig. 3c). We therefore propose that the C-terminal domains of C7 may position the C5b C345C domain in an activated conformation that enables the recruitment of complement proteins to the MG scaffold.

**Structural transitions of MACPF domains.** The MACPF domain of complement proteins undergoes dramatic structural rearrangements during pore formation. Structures of soluble[17,31–33] and membrane inserted forms[16] of complement proteins have shown that two helical bundles within the MACPF (TMH1, TMH2) unfurl to form transmembrane β-hairpins. To understand how the helix-to-hairpin transition of MACPF

residues is mediated, we sought to model a conformation of C9 for which MAC assembly is stalled. As the density for C9 is best resolved in the 2C9-sMAC complex, we used this map for subsequent analyses. We next subtracted density for C5b6, C7, and C8 from the raw images. By then focusing refinement on the remaining C9, we were able to generate a map at 3.3 Å resolution in which density for both copies of C9 were clearly resolved (Fig. 4 and Supplementary Fig. 1).

We discover that within sMAC, C9 adopts two distinct conformations. Upon binding C8, the first C9 molecule undergoes a complete transition in which both TMH1 and TMH2 are extended, and CH3 is in a position analogous to MAC (Supplementary Fig. 7). By contrast, only one helical bundle (TMH1) has unfurled in the terminal C9 conformation. There is no density for the extended hairpins of TMH2; instead, these residues adopt a conformation similar to that of soluble C9 (Fig. 4a, b). To validate our model for the terminal C9 conformation, we used the XL-MS data and mapped intramolecular C9 cross-links. We identify four cross-links within TMH2 that satisfy all the distance restraints when plotted on the stalled conformation (9 Å, 14 Å, 17 Å, 20 Å) (Fig. 4e). Two are overlength when mapped onto the MAC conformation of C9 (17 Å, 28 Å, 32 Å, 59 Å), further supporting that C9 transmembrane hairpins unfurl sequentially. In the stalled C9, CH3 follows the movement of the central β-sheet of the MACPF domain, which aligns with the preceding monomer (Fig. 4b). In doing so, the helical TMH2 bundle swings out and positions the side chain of $Arg_{348}$ proximal to a glycan (NAG-$Asn_{394}$) on the β-strand of the penultimate C9 molecule (Fig. 4c). We hypothesize that this interaction may play a role in stabilizing an intermediate state in which the preceding monomer templates TMH1 β-strands and correctly positions TMH2 for sequential unfurling (Fig. 4f).

We next considered how this stalled conformation of C9 might help us understand disease-related variants. Several point mutations within the C9 MACPF are associated with Age-related Macular Degeneration (AMD)[34]. In particular, substitution of $Pro_{146}$ (P146S) within a loop of C9 directly influences polymerization of C9. Unlike its position in the MAC, we find that this loop flips outward in the stalled conformation, sterically blocking subsequent C9 incorporation (Fig. 4d). Substitution of proline at this position may impact the ability of this loop to serve as a checkpoint for polymerization. Our structural model is further supported by mutational studies showing that mutation of $Pro_{146}$ increases C9 polymerization[35]. Taken together, our sMAC model provides a structural timeline in which the sequential extension of transmembrane β-hairpins may be regulated by the preceding monomer and a proline latch on the polymerizing MACPF interface (Fig. 4f).

## Discussion

In sMAC, fluid-phase chaperones prevent bystander damage by trapping MACPF-containing complement complexes. As such, sMAC represents an important model system to probe structural transitions of pore-forming proteins. In addition to MAC, two other human immune pores (perforin-1 and MPEG-1) rupture membranes using MACPF domains (Supplementary Fig. 7). While structures are available for both soluble and membrane-inserted states of these complexes[36–38], mechanistic details governing the transition between the two conformations have been more challenging to study.

Here we capture a stalled conformation of C9 that allows us to define a structural pathway for MACPF pore-formation (Fig. 4f). In our structure of sMAC, TMH residues of complement proteins are stably unfurled into their β-hairpin conformation even in the absence of a lipid bilayer. Therefore, it may be possible that an arc

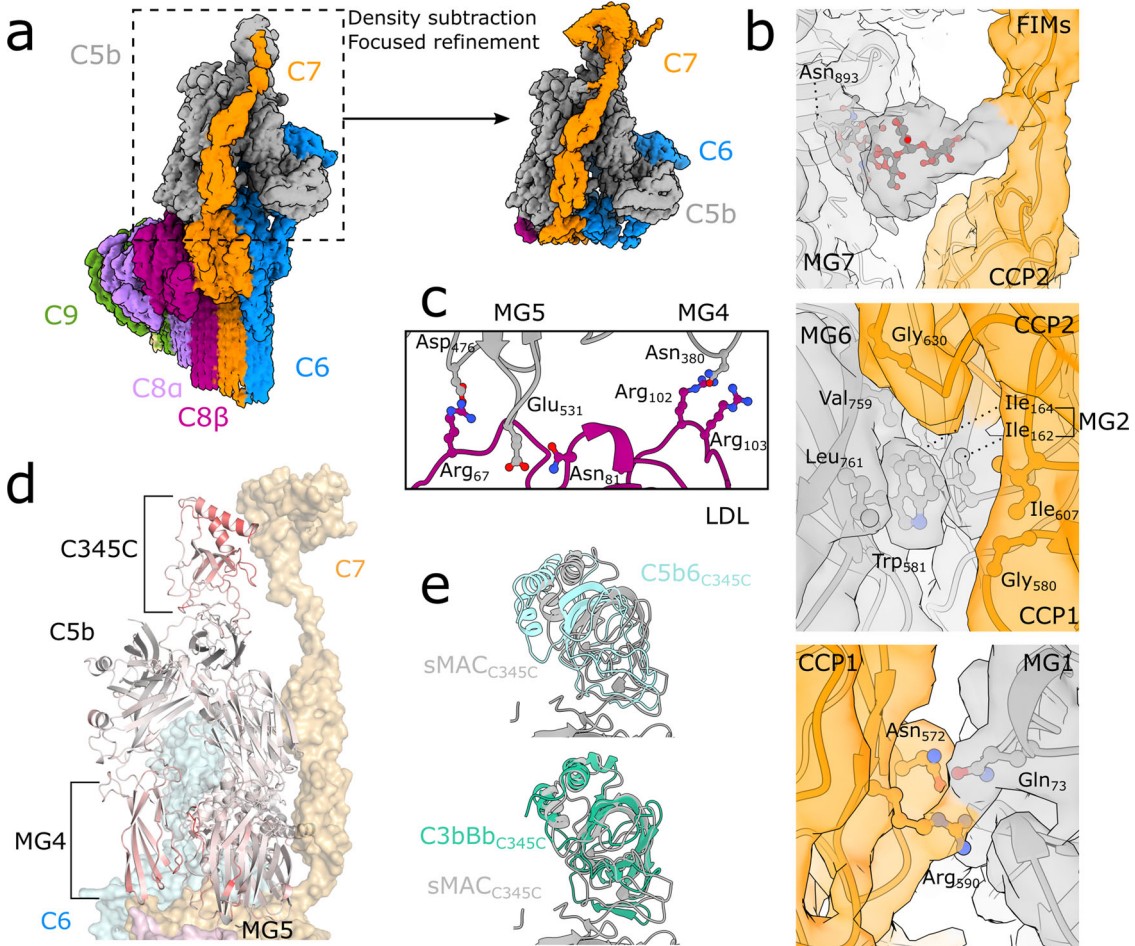

**Fig. 3 C7 connects conformational changes of the C345C domain with the MG scaffold of C5b. a** CryoEM reconstruction of 2C9-sMAC highlighting the region used in subsequent focused refinements (black-dotted lines). Inset corresponds to the map resulting from density subtraction of the MACPF-arc and focused refinement on C5b. Density is colored by protein composition. **b** Panels show density from the focused refined map (colored by protein composition) overlaid with the sMAC atomic model (ribbons) at three interaction interfaces between C7 (orange) and C5b (gray). Glycan extending from C5b:Asn$_{893}$ stabilizes a linker between the C-terminal CCP and first FIM domain of C7 (top). C5b:Tryp$_{581}$ locks into a hydrophobic hinge between the CCP1 and CCP2 domains of C7 (middle). Ionic interactions between the first CCP of C7 and the MG1 domain of C5b (bottom). Side-chains of interface residues shown as sticks. **c** Interface between the C8β LDL domain (purple) and C5b MG scaffold (MG4 and MG5 in grey). Side-chains of interface residues are shown as sticks. **d** C5b within sMAC (ribbons) colored by RMSD with superposed C5b from the C5b6 crystal structure (PDB ID: 4A5W). Red indicates residues with maximal differences. C345C, MG4 and MG5 domains of C5b are highlighted. C6 (blue), C8 (pink) and C7 (orange) are shown as semi-transparent surfaces for reference. **e** Superposition of C5b within sMAC (grey) with corresponding residues in the soluble C5b6 complex (cyan) (PDB ID: 4A5W) showing movement of the C345C domain (top panel). Superposition of C5b from sMAC (grey) with the structural homolog C3b (green) from the C3b:Bb:Properdin complex (PDB ID: 6RUR). In both panels, alignments were done on the full molecule and C345C domains were cropped for clarity.

comprised of C6, C7, C8, and up to 3 copies of C9 could serve as a minimal nucleation unit that inserts into the membrane to initiate MAC pore formation. Indeed, bacterial homologs of the cholesterol-dependent cytolysin superfamily can form oligomeric arcs that remodel lipid bilayers, generating membrane-inserted assembly intermediates that are non-conductive[39]. To complete the MAC pore, further C9 molecules could then be sequentially inserted by templating leading edge β-strands, reminiscent to the folding of β-sheet porins by the β-barrel assembly machine (BAM)[40].

As soluble monomers bind the MAC precursor, the central kinked β-sheet of the MACPF straightens to align with the pre-ceding monomer. Our data show how two helical bundles (CH3 and TMH2) rotate to stabilize a transition state, in which a basic residue on TMH2 interacts with a glycan on the β-strands of a preceding monomer. We hypothesize that this interaction may play a role in correctly positioning TMH2 as the first β-hairpin (TMH1) geometry is templated by the strands of the preceding

monomer. Indeed, removal of N-linked glycans from C9 resulted in MACs with distorted β-barrels[16]. The pore is then propagated by the sequential insertion of the second β-hairpin, TMH2. We note that within the pore conformation, MPEG-1 contains a glycan on the leading edge of TMH2[38], homologous to C9. Remarkably, superposition of a soluble conformation of MPEG-1 positions a basic residue proximal to a TMH2 glycan of the preceding monomer (Supplementary Fig. 7). We therefore pro-pose that glycans may play a role in stabilizing transition states of MACPF pore-formation. More broadly, glycans of MACPF-containing proteins are shown to have a structural role in astrotactin-2[41] and are functional regulators of perforin-1 activity[42].

In addition to understanding immune pore-formation, our combined cryoEM and XL-MS data provide a molecular basis for how clusterin binds cargo. Based on results presented here, we show that clusterin is a predominantly helical protein that binds to a negatively charged surface on sMAC. Specifically, we find

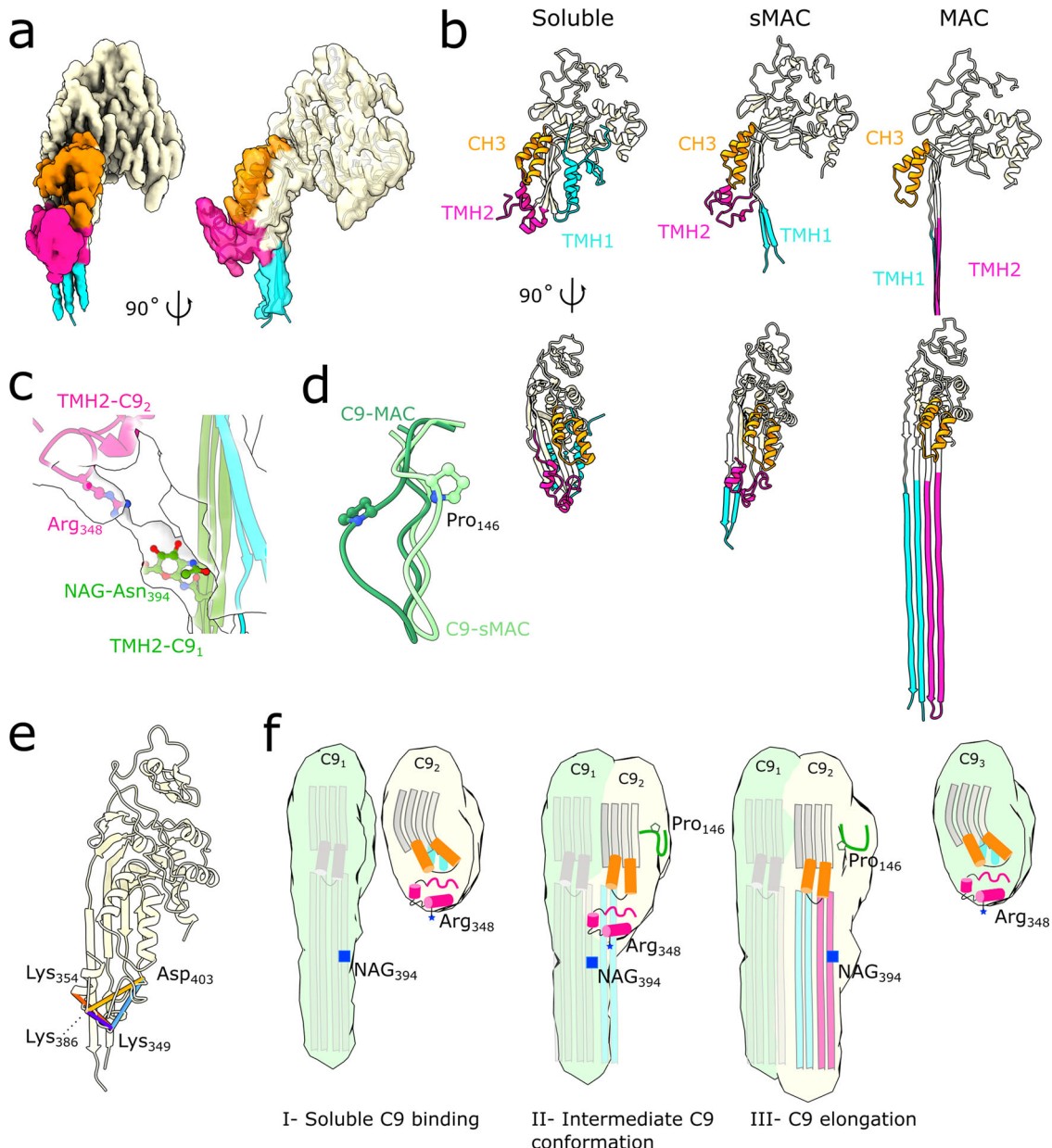

**Fig. 4 sMAC traps an alternative conformation of C9. a** Density for the terminal C9 in the 2C9-sMAC reconstruction (EMD-12647). Density corresponding to the CH3 (orange), TMH2 (pink) and TMH1 (cyan) regions of the MACPF domain are highlighted. The remainder of C9 is beige. Model for the alternative C9 conformation is overlaid (ribbon). **b** Ribbon diagrams for three conformations of C9: soluble C9 from the murine crystal structure (PDB ID: 6CXO) left panel, terminal C9 in 2C9-sMAC, and transmembrane conformation of C9 in MAC (PDB ID: 6H03). CH3, TMH1, and TMH2 regions of the MACPF domain are colored as in a. For the MAC conformation, the full length of C9 TMH hairpins are shown in the bottom right panel. **c** Interaction between the penultimate C9 TMH2 β-hairpins (green) with the helical TMH2 of the terminal C9 (pink). Side-chains of interface residues (NAG-Asn$_{394}$ and Arg$_{348}$) are shown as sticks. TMH1 of the terminal C9 (cyan) is shown for context. Density for this region is shown as a transparent surface. **d** Superposition of the C9 MACPF from MAC (dark green) and the conformation in the terminal C9 (light green) shows differences in a loop that contains the disease affected residue Pro$_{146}$ (P146S). **e** Cross-links between TMH2 residues (Lys$_{354}$, Lys$_{386}$, Lys$_{349}$, Asp$_{403}$) derived from XL-MS and mapped on the terminal C9 conformation of 2C9-sMAC (beige ribbons). Distance lengths: 9 Å (purple), 14 Å (red), 17 Å (blue), 20 Å (yellow) are shown. Source data are provided as a Source Data file. **f** Schematic showing structural timeline of MACPF pore formation. In the soluble conformation (C9$_2$, C9$_3$), the MACPF domain comprises a central kinked β-sheet (grey) and the pore-forming residues within TMH1 (cyan) and TMH2 (pink) are helical. Upon binding the leading edge of the oligomer (C9$_1$), the central β-sheet straightens and two other MACPF regions: CH3 (orange cylinders) and a proline loop (dark green) rotate as TMH1 (cyan) is released. The intermediate conformation is stabilized by the interaction between a basic residue on TMH2 (Arg$_{348}$, blue star) with a glycan on the β-strands of the preceding monomer (NAG-Asn$_{394}$, blue square). This interaction may play a role in positioning TMH2 before the hairpins are sequentially released to propagate the pore.

that clusterin caps the polymerizing face of the leading MACPF domain, thus providing a mechanism for how clusterin prevents C9 polymerization[43]. We also define a binding site for clusterin that spans LDL domains of multiple complement proteins, in agreement with studies that suggest clusterin binds a structural motif common to C7, C8α, and C9[43]. Density for clusterin at this site is the most well resolved in the 3C9-sMAC maps; therefore, the interaction is likely secured through repeated contacts across oligomers. Clusterin also binds oligomeric amyloid-beta fibrils through the same interaction interface as sMAC[44]. As clusterin inhibits amyloid-beta aggregation in vitro[45,46], this may serve as a general mechanism for how clusterin blocks polymerization of potentially pathogenic proteins. In addition to the LDL binding site, we observe a second stretch of density below the EGF domains of C9. This density also maps to a clusterin binding-site supported by our XL-MS data. Indeed, our MS data reveal that multiple copies of clusterin are present in sMAC. While the role of this second binding site in regulating pore formation remains unclear, it may be important in linking cargo with an endocytic receptor for clearance[47]. Clusterin is rapidly emerging as a key player at the crossways between clearance pathways and immune homeostasis[48]. Our structure of sMAC shows how clusterin can bind polymerizing cargo and opens new lines of investigation into the role of clusterin in immunobiology.

In summary, we have solved multiple structures of complement activation macromolecules by combining cryoEM and mass spectrometry. These structures underpin a complement control pathway that prevents bystander damage during an immune response. Our structural analyses show how chaperones trap pore-forming intermediates and bind oligomeric proteins to prevent further polymerization, which may also be relevant for controlling pathogenic aggregation of amyloids. Finally, we anticipate that our structural findings will provide mechanistic insight into transition states of immune pores.

## Methods

**CryoEM sample and grid preparation.** To prepare cryoEM grids, sMAC (Complement Technologies) provided in 10 mM sodium phosphate, 145 mM NaCl, pH 7.3 was diluted to 0.065 mg/ml in 120 mM NaCl, 10 mM Hepes pH 7.4 and used for freezing within the hour. In all, 4 μl were deposited on glow-discharged gold grids with a lacey carbon film (Agar Scientific). After a 10 s incubation at room temperature and 95% humidity, grids were blotted and flash frozen in liquid ethane using a Vitrobot Mark III (Thermo Fisher Scientific). CryoEM conditions were screened using a Tecnai T12 (Thermo Fisher Scientific) operated at 120 kV. Two data sets were collected using EPU version 1.12.079 on 300 kV Titan Krios microscopes (Thermo Fisher Scientific) equipped with K2 Quantum direct electron detectors (Gatan) operated in counting mode at a magnification of 130k, corresponding to a calibrated pixel size of 1.047 Å and 1.048 Å for each dataset, respectively. The first data set was collected at 0 degree tilt and consisted of 11,107 image stacks taken over a defocus range of −1.1 to −2.3 μm in 0.3 μm steps. The total exposure time was 8 s which included 40 frames and resulted in an accumulated dose of 40 electrons per Å². As initial processing of the first dataset showed the particles adopted a favored orientation, we used the cryoEF software[49] to estimate the appropriate tilt angle for data collection and acquisition of missing views. The second data set was collected at a 37° tilt and consisted of 2596 image stacks over a defocus range of −1.1 to −2.1 μm. Movie stacks were collected with similar conditions as dataset-1. A summary of the imaging conditions is presented in Supplementary Table 1.

**Image processing.** Electron micrograph movie frames were aligned with a Relion-3.1[50] implementation of MotionCor2. CTF parameters were estimated using CTFFIND4-1[51]. The datasets were manually curated to remove movies with substantial drift and crystalline ice. For dataset-1, a small subset of micrographs across the whole defocus range were randomly selected for manual picking in Relion. Following 2D classification, classes with diverse orientations were used for Autopick of the entire dataset. Due to the use of a lacey carbon film grid, a large portion of micrographs were acquired over a steep ice gradient and with visible carbon edges. Relion Autopick performed best to avoid over-picking on carbon when calibrated to pick in thinner ice areas. To complement the particle stacks, crYOLO[52] was used in parallel and specifically trained to pick in thicker ice areas. Particles were extracted at 4.188 Å/px (bin by 4) and subjected to iterative 2D classification to remove ice contamination, carbon edges and broken particles. An

initial model was generated in Relion which was strongly low-pass filtered to 60 Å resolution and used as a starting model for 3D auto-refinement. The initial refinement with 595,890 particles revealed strong particle distribution anisotropy. To improve the diversity of particle orientations, projections of low occurrence views were generated from the initial reconstruction and used as templates for re-picking the micrographs in Relion. In parallel, the reweight_particle_stack.py script (available on the Leschziner lab Github) was iteratively used to randomly select and remove particles from the over-represented orientation, followed by 3D auto-refinements of the remaining particles. Duplicated particles identified within a distance threshold of 10 nm were removed at each data merging step resulting in a final 389,625 particles in dataset-1. For dataset-2, manual picking of a small subset of micrographs followed by 2D classification at 4.192 Å/px (bin by 4) was first done and five classes were then used for autopicking of all the micrographs in Relion. The picks were cleaned in iterative 2D classifications, resulting in a final 83,376 particles. Unbinned particles from dataset-1 and -2 were then merged for a 3D auto-refinement yielding a consensus map at 3.8 Å with 473,001 particles. Low map quality in the C9 arc suggested heterogeneity in this area. The aligned particles were thus subtracted to only keep the C9 arc, followed by 3D classification without refinement (bin 4, $T = 20$, 10 classes) which identified three sMAC stoichiometries, with 1 C9 (20.6 %), 2 C9 (29.5 %), or 3 C9 (17.9 % of particles). Particle stacks were reverted to the original images and each class was 3D auto-refined individually, followed by Bayesian polishing and multiple rounds of per-particle CTF refinements[53] (Supplementary Fig. 1) to generate the final reconstructions for 1C9-sMAC (EMD-12649), 2C9-sMAC (EMD-12651), and 3C9-sMAC (EMD-12650) (resolutions 3.8 Å, 3.3 Å, and 3.5 Å respectively). Further heterogeneity analysis via multi-body refinement of sMAC revealed flexibility within the complex. To better resolve the C-terminal domains of C5b and C7, density subtraction of the MACPF arc was done on the best-resolved 2C9-sMAC map. The subtracted particles were then aligned in a masked 3D auto-refinement to generate a reconstruction at 3.6 Å (EMD-12648). Similarly, the 2C9-sMAC particles were density subtracted only keeping the signal from the terminal 2 C9s. The 2C9 subtracted particle set was further subjected to 2D classification to remove any remaining heterogeneity in the number of C9s. A final 96,118 particles were selected for a masked 3D auto-refinement resulting in a 3.3 Å reconstruction (EMD-12647). The 3C9-sMAC reconstruction contained weak density above the C9 arc suggesting poor alignment of particles in this area. To better resolve this area, density was subtracted from the raw particles using a generous mask extending above the density and containing the three terminal C9s, followed by 3D auto-refinement. A second round of subtraction was done on the newly aligned particles to only keep the MACPF region and the newly resolved density above it. A final masked 3D auto-refinement generated a reconstruction at 3.8 Å (EMD-12646). Resolutions of maps were determined using the masking-effect corrected Fourier Shell Correlation (FSC) as implemented in Relion post-processing.

**Model building and refinement.** Models were built and refined into locally sharpened maps generated by DeepEMhancer[54]. To create initial starting models, complement proteins (C5b, C6, C7, and C8) from MAC (PDB ID: 6H04)[16] were augmented or substituted with higher resolution structures for individual domains. Specifically, the TED, MG8, and C345C domains of C5b and C6 CCP domains were derived from the C5b6 crystal structure (PDB ID: 4A5W)[14]; C7-FIM domains were included from the NMR structure (PDB ID: 2WCY)[55]. As we observe two conformations of C9 in sMAC, we created two unique starting models (C9$_1$ and C9$_2$). C9$_1$ was derived from its MAC conformation (trimmed to remove unresolved TMH2 residues). C9$_2$ was generated by replacing TMH2 residues from the MAC conformation of C9 (PDB ID: 6H03)[16] with TMH2 helices from the soluble mouse C9 structure (6CXO)[17]. Amino acids from the murine model were subsequently changed to the human sequence in Coot[56]. All missing amino acids in C9 were manually built in areas where density was present. Flexible regions of TMH1 hairpins were removed to match the density. The terminal C9 EGF domain was also removed from the model as density for this region was not well defined. Where merited by the EM density, missing loops, and individual amino acids were manually built in Coot. Where appropriate, known calcium ions were also added in Coot. All models were refined into the EM densities using ISOLDE[57] as a built-in module in ChimeraX[58], with secondary structure geometries restraints and ligand position restraints applied. All reasonable disulfides were formed in ISOLDE to stabilize the protein chains during refinements.

We used density subtracted focus-refined maps to generate more accurate models for individual subregions. To generate the model for the interaction interface between C5b and C7, individual domains derived from higher resolution structures (C5b:C345C, C5b:MG8, C6:CCPs, and C7:FIMs) were rigid body fitted into the density subtracted C5b-focus refined 2C9-sMAC (EMD-12648) using ChimeraX. Models were merged with the rest of C5b, C6, and C7 in Coot and further refined in ISOLDE where adaptive distance restraints were kept active for C5b:C345C and two C7:FIM domains. The density subtracted C9-focus refined 2C9-sMAC (EMD-12647) was used to refine atomic models for the two C9 conformations using ISOLDE.

Following building and refinement in the density subtracted maps, final composite models were created for both 2C9-sMAC (PDB ID: 7NYD) and 3C9-sMAC (PDB ID: 7NYC). To generate the composite 2C9-sMAC model we merged models derived from the C5b-focus refined map and those derived from the

C9-focus refined map with remaining domains of complement proteins. The composite model was then refined into the full 2C9-sMAC map (EMD-12651), focusing on interaction interfaces between protein chains. Here the whole of the terminal C9 (C9₂) was stabilized with adaptive distance restraints and not refined as the density for this chain was of poorer quality in the full 2C9-sMAC map. To build the 3C9-sMAC model, C9₁ from 2C9-sMAC was duplicated and fitted in the C9₁ and C9₂ positions of 3C9-sMAC, while the terminal C9 from 2C9-sMAC was placed in the last position of the 3C9-sMAC arc (C9₃) in ChimeraX. All other sMAC components were then added and the composite model was refined in the 3C9-sMAC map (EMD-12560) using ISOLDE, with secondary structure geometries restraints and ligand position restraints applied. Again, adaptive distance restraints were imposed for the C5b:C345C domain. In both 2C9- and 3C9-sMAC models, side chains of the C345C domain were removed as the resolution did not permit confident refinement of their positions. In addition, side chains from other places across the model were punctually removed after refinement where density was lacking. N-linked glycans were built using the Carbohydrate tool in Coot. For C-linked and O-linked glycans, the sugars were fitted in the density from the Coot monomer library and linked to the protein chain using the Acedrg tool[59]. Glycans and linked side chains were then real-space refined in Coot. Finally, local B-factors of the composite models were refined in REFMAC5[60] using the local resolution filtered map from Relion. Map-Model FSC and the overall quality of the models were assessed in the full 2C9- and 3C9-sMAC maps using the cryoEM validation tools in Phenix[61] and MolProbity[62].

**Map visualization and analysis**. Density maps and models were visualized in ChimeraX. Local resolution of the maps and angular distribution of the particles were assessed in Relion and visualized in ChimeraX. Coulombic electrostatic potentials of interaction interfaces were calculated and visualized in ChimeraX. Interaction interfaces and structural rearrangements of complement proteins were analyzed in Coot and ChimeraX. RMSD values between structures of C5b6 were calculated in PyMOL Molecular Graphics System, Version 2.0 Schrödinger, LLC. Figures were generated in ChimeraX, PyMOL, and DataGraph.

**Bottom-up LC-MS/MS analysis of sMAC**. For bottom-up LC-MS/MS analysis, purified sMAC (Complement Technologies) in PBS buffer (10 mM sodium phosphate, 145 mM NaCl, pH 7.3) at a concentration of 1 mg/ml were introduced into the digestion buffer containing 100 mM Tris-HCl (pH 8.5), 1% w/v sodium deoxycholate (SDC), 5 mM Tris (2-carboxyethyl) phosphine hydrochloride (TCEP), and 30 mM chloroacetamide (CAA). Proteins were digested overnight with trypsin at an enzyme-to-protein-ratio of 1:100 (w/w) at 37 °C. After, the SDC was precipitated by bringing the sample to 1% trifluoroacetic acid (TFA). The supernatant was collected for subsequent desalting by an Oasis µElution HLB 96-well plate (Waters) positioned on a vacuum manifold. The desalted proteolytic digest was dried with a SpeedVac apparatus and stored at −20 °C. Prior LC-MS/MS analysis, the sample was reconstituted in 2% formic acid (FA). Approximately 300 fmol of peptides was separated and analyzed using the HPLC system (Agilent Technologies) coupled on-line to an Orbitrap Fusion Lumos mass spectrometer (Thermo Fisher Scientific). The peptides were first trapped on a 100 µm × 20 mm trap column (in-house packed with ReproSil-Pur C18-AQ, 3 µm) (Dr. Maisch GmbH, Ammerbuch-Entringen, Germany) and then separated on the in-tandem connected 50 µm × 500 mm analytical column (in-house packed with Poroshell 120 EC-C18, 2.7 µm) (Agilent Technologies). Mobile-phase solvent A consisted of 0.1% FA in water, and mobile-phase solvent B consisted of 0.1% FA in acetonitrile (ACN). The flow rate was set to 300 nL/min. A 90 min gradient was used as follows: 0–5 min, 100% solvent A; 13–44% solvent B within 65 min; 44–100% solvent B within 3 min; 100% solvent B for 5 min; and 100% solvent A for 12 min. The mass spectrometer was operated in positive ion mode, and the spectra were acquired in the data-dependent acquisition mode. A Nanospray was achieved using a coated fused silica emitter (New Objective) (outer diameter: 360 µm; inner diameter, 20 µm; tip inner diameter, 10 µm) biased to 2 kV. For the MS scans, the mass range was set from 350 to 1800 $m/z$ at a resolution of 60,000, maximum injection time 50 ms, and the normalized automatic Gain Control (AGC) target set to $4 × 10^5$. For the MS/MS measurements, higher-energy collision dissociation (EThcD) with supplementary activation energy of 27% was used. MS/MS scans were performed with fixed first mass 100 $m/z$. The resolution was set to 30,000; the AGC target was set to $1 × 10^5$ the precursor isolation width was 1.6 Da and the maximum injection time was set to 250 ms. The LC-MS/MS data were searched against the UniProtKB/Swiss-Prot human proteome sequence database with MaxQuant software (version 1.5.3.30)[63] with the standard settings and trypsin as digestion enzyme (Supplementary Data file). For label-free quantification intensity based absolute quantification (iBAQ) values were selected as output.

**Cross-linking**. Purified sMAC (10 µg, Complement Technologies) was cross-linked using 0–2 mM disuccinimidyl sulfoxide (DSS) or 0–20 mM 4-(4,6-Dimethoxy-1,3,5-triazin-2-yl)-4-methylmorpholinium chloride (DMTMM) for 30 min at RT, followed by quenching using a final concentration of 50 mM Tris. Cross-linked samples were analyzed by sodium dodecyl sulfate polyacrylamide gel electrophoresis (SDS-PAGE) and blue native-PAGE (BN-PAGE) to determine an optimal cross-linker to protein ratio. The optimal DSS and DMTMM concentration (1 mM and 15 mM, Supplementary Fig. 4) was used for cross-linking of 20 µg sMAC (0.5 mg/ml) in triplicates. After quenching the reactions, protein precipitation was performed by adding three times 55 µl cold acetone and subsequent incubation at −20 °C overnight. Precipitated samples were centrifuged at 12,000×g for 20 min. After careful removal of the supernatant, the remaining pellet was air-dried until no acetone solution was visible anymore. Pellets were resuspended in 50 µl ammonium bicarbonate with 0.33 µg trypsin (1:60) and incubated with shaking for 4 h at 37 °C. The solubilized pellets were reduced by 5 mM TCEP for 5 min at 95 °C followed by alkylation with 30 mM CAA for 30 min at 37 °C. Digestion was performed overnight by 0.4 µg trypsin (1:50) at 37 °C. The samples were deglycosylated using PNGase F (1 unit/10 µg) for 3 h at 37 °C. Next, the samples were acidified with TFA before desalting using an Oasis HLB plate (Waters, Wexford, Ireland). Finally, the eluent was dried completely and solubilized in 2% FA before LC-MS/MS-analysis.

**LC-MS/MS analysis of cross-linked sMAC**. Data was acquired using an Ultimate 3000 system (Thermo Scientific) coupled on-line to an Orbitrap Fusion (Thermo Scientific) controlled by Thermo Scientific Xcalibur software. First, peptides were trapped using a 0.3 × 5 mm PepMap-100 C18 pre-column (Thermo Scientific) of 5 µm particle size and 100 Å pore size prior to separation on an analytical column (50 cm of length, 75 µm inner diameter; packed in-house with Poroshell 120 EC-C18, 2.7 µm). Trapping of peptides was performed for 1 min in 9% solvent A (0.1 % FA) at a flow rate of 0.03 ml/min. The peptides were subsequently separated as follows: 9–13% solvent B (0.1% FA in 80% CAN) in 1 min, 13–44 % in 70 min, 44–99% in 3 min, and finally 99% for 4 min. The flow was 300 nl/min. The mass spectrometer was operated in a data-dependent mode. Full-scan MS spectra from 350–1800 Th were acquired in the Orbitrap at a resolution of 120,000 with the AGC target set to $1 × 10^6$ and maximum injection time of 100 ms. In-source fragmentation was turned on and set to 15 eV. Cycle time for MS² fragmentation scans was set to 2 s. Only peptides with charge states 3–8 were fragmented, and dynamic exclusion properties were set to $n = 1$, for a duration of 20 s. Fragmentation was performed using in a stepped HCD collision energy mode (27, 30, 33%) in the ion trap and acquired in the Orbitrap at a resolution of 50,000 after accumulating a target value of $1 × 10^5$ with an isolation window of 1.4 Th and maximum injection time of 180 ms.

**Data analysis of XL-MS**. The raw data was first searched using MaxQuant (version 1.6.10.0) to generate a database for the cross-linking search. The signal peptides of the sMAC components were removed. Next, the raw data was searched using pLink (version 2.3.9)[64] using the conventional cross-linking flow type and DSS or DMTMM as cross-linker (Supplementary Data file). Trypsin was set as a digestion enzyme with two missed cleavages. The peptide mass was 600–6000 Da and peptide length 6–60 amino acids. Carbamidomethyl (C) was set as fixed modification and oxidation (M), and acetyl (protein N-term) as variable modifications. Furthermore, we added hex (W) and sulfo (Y) as variable modifications to account for c-mannosylation of thrombospondin type 1 domains and sulfation of vitronectin. Precursor, fragment, and filter tolerance were set to 10 ppm and FDR 5% at the PSM level. Cross-linked and loop-linked sites identified in all triplicates were selected for further analysis (DSS: 319 XL, DMTMM: 221 XL, Source data file). The cross-links were mapped on the sMAC structures using PyMOL to obtain Cα-Cα distances. Distance restraints were set to <20 Å for DMTMM and <30 Å for DSS cross-links. The circos plots were generated in RStudio and only cross-links involving the complement components C5, C6, C7, C8α, C8β, C8γ, and C9 were included.

**Mass photometry**. Mass photometry data was collected on a Refeyn One^MP instrument using ReFeyn Acquire^MP software. The instrument was calibrated with a protein standard (made in-house). The following masses were used to generate a standard calibration curve: 73, 149, 479, and 800 kDa. Borosilicate coverslips were extensively cleaned with Milli-Q water and isopropanol prior to the measurements. sMAC (5 µl) was applied to 10 µl buffer (10 mM sodium phosphate, 145 mM NaCl, pH 7.3) on a coverslip resulting in a final concentration 13 µg/ml. Movies were acquired by using ReFeyn Acquire^MP software for 6000 s with a frame rate of 100 Hz. The particle landing events detected were 5480 (5241 binding and 239 unbinding). All data was processed in ReFeyn Discover^MP software. Masses of sMAC complexes were estimated by fitting a Kernel density distribution to the landing events. Gaussian fit to the mass of GroEL from the protein standard (800 kDa) was generated for peak width reference.

**Modeling of the clusterin core**. Initial clusterin models were generated using trRosetta[26] with the clusterin protein sequence as input. Alignment of the five generated models in PyMOL revealed a common core composed of residues 40–124 and 256–427. The three models with the best RMSD were selected for further modeling and used as input for comparative modeling using Modeller[29]. The five disulfide bonds were included as restraints for the modeling. The model with the lowest DOPE score was selected as the final model. The core of the model was verified by mapping the clusterin intra-links on the structure using PyMOL to obtain Cα-Cα distances.

**Reporting summary**. Further information on research design is available in the Nature Research Reporting Summary linked to this article.

## Data availability

Data supporting the findings of this manuscript are available from the corresponding authors upon reasonable request. A reporting summary for this article is available as a Supplementary Information file. Source data are provided with this paper. The MS raw data generated in this study have been deposited in the ProteomeXchange partner MassIVE database under the accession code MSV000087092. The processed MS data generated in this study underlying (Figs. 1c, 2b, and 4e and Supplementary Figs. 3b–d, 4c–f, and 6a–b) are included as a Source Data file. The cryo EM maps generated in this study have been deposited in the Electron Microscopy Data Bank under the accession codes EMD-12646, EMD-12647, EMD-12648, EMD-12649, EMD-12650, and EMD-12651. The structural models generated in this study have been deposited in the Protein Data Bank under the accession codes 7NYC and 7NYD. Structural models used to initiate model building were accessed from the Protein Data Bank under the accession codes 6H03, 6H04, 4A5W, 2WCY, and 6CXO. Structural models used in data analysis were accessed from the Protein Data Bank under the accession codes 1W33, 6SB3, 6SB5, 3NSJ, 4OEJ, and 5J68. Source data are provided with this paper.

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

## Acknowledgements

We thank S. Islam for computational support and P. Simpson for EM support. Initial screening of samples was carried out at Imperial College London Centre for Structural Biology; cryoEM data was collected at Diamond Light Source. We thank Diamond for access and support of the Cryo-EM facilities at the UK national electron bio-imaging centre (eBIC), proposal EM18659, funded by the Wellcome Trust, MRC, and BBSRC. This project has received funding from the European Research Council (ERC) under the European Union's Horizon 2020 research and innovation program (grant agreement No. 864751) to D.B.; E.C.C. is supported by the CRUK Convergence Science Centre at Imperial College London (C24523/A26234) and the EPSRC Centre for Doctoral Training: Chemical Biology: Physical Sciences Innovation (EP/L015498/1). M.V.L., V.F., and A.J.R.H. acknowledge support from the Netherlands Organization for Scientific Research (NWO) funding the Netherlands Proteomics Centre through the X-omics Road Map program (project 184.034.019) and the ENW-PPP project 741.018.201, and the EU Horizon 2020 program INFRAIA project Epic-XS (Project 823839). M.V.L. further acknowledges fellowship support from the Independent Research Fund Denmark (Project 9036-00007B).

## Author contributions

A.M. conducted cryoEM work. A.M. and E.C.C. built and refined atomic models of complement proteins. A.M. and M.V.L. generated structural models for clusterin. M.V.L. and V.F. performed mass spectrometry experiments. A.M., D.B., V.F. and A.J.R.H. conceived the ideas. A.M. and D.B. analyzed cryoEM data. M.V.L., V.F. and A.J.R.H. analyzed mass spectrometry data. D.B. wrote the manuscript. A.M. and M.V.L. generated the figures. All authors assisted with manuscript editing.

## Competing interests

The authors declare no competing interests.
