## [Peer Review File · Nature Communications]

Structural basis of soluble membrane attack complex packaging for clearanceREVIEWER COMMENTS

Reviewer #1 (Remarks to the Author):

This is a good paper and it deserves to be published, after appropriate revision. The paper describes an important and insight-providing set of structures of the sMAC which significantly enhance our understanding of MAC biology and assembly.

A key starting point is to recognise that this paper concerns "an immune activation complex that is formed from MAC assembly precursors released into plasma and scavenged by blood-based chaperones". Thus we are dealing both with a complex which exists in serum and which is of relevance to the initiation of complement membrane attack, and with a complex which at the same time reveals details of relevance to piecemeal assembly of the MAC on a target membrane, one subunit at a time. That in any case is what I took from the paper as a whole, though I felt the emphasis was initially on the first aspect - membrane attack by sMAC/its prevention by vitronectin and clusterin - and only latterly on the insights provided for on-the-membrane complex assembly. Clearly, the particular focus on this first aspect may relate to the clinical relevance of the study - e.g. to ARMD - but I guess as with any structural analysis whatever the results there is likely to be something of relevance to more than one aspect of the system's functioning. In this case, most strikingly, to the structural stability of MAC complexes (sMACs) with their hydrophobic beta hairpins deployed but which nevertheless are (i) free in solution but (ii) can partition into a target bilayer (because why else the need for clusterin and vitronectin?).

Some specific comments:

1. The densities for clusterin and vitronectin prove hard to define. I don't see a worrisome issue here - see below - but (page 5, end of para. 1) I wondered if some 2d classification of negatively stained images could give a bit of an insight into the different states adopted by the bound clusterin and vitronectin subunits? This may indicate if there are a relatively small number of statically disordered states or a larger and potentially undefinable set of dynamic conformations. If static then the different states may have some functional relevance.

2. Be clear in the figure legend 2d that the clusterin depiction is based on a homology model as described on page 8 and in the Methods, and not on an experimental structure.

3. A critical thing to address is the unfurling of the beta hairpins in solution. Generally they are considered in MACPF/CDC proteins to unfurl on membranes - here in the structure determined they are already unfurled (though only partly in the case of the second copy of C9). This suggests that the MAC can insert in an already partly pore-forming state (an arc of C5b-9 up to C5b-999) and it then extends in size in the membrane with the recruitment of additional copies of C9. This is an important observation and entirely feasible. A good comparator would be the likely spontaneous insertion (at least according to the assisted model) of small (I would argue) beta sheeted porins by the BAM complex. [In my view of the two models for BAM the templating model might likely apply to larger barrels, the assisted model to smaller ones.] Another would be the work of Mark Wallace and colleagues on alpha-hemolysin which suggests ways in which membranes remodel themselves when confronted with inserting beta barrel structures.

It seems to me that the paper discusses the insights more in terms of what would happen with piecemeal assembly of the MAC on the target membrane, but the actual structure determined relates to a different scenario. It casts light on both processes - step by step and insertion at once of C5b-9n where n=1-3 - but the relevance to both should I think be teased out more, with comment on the striking stability of the unfurled TMHs for C6-C9 included.

4. On page 12 the role of glycans is discussed and the relationship between the glycans found in perforin-2/MPEG1 and C9 is described as "analogous". Is not the relationship specifically

"homologous"? In general, the proposal of importance for the glycans in MAC structure/mechanism is insightful and welcome - on this point it may be worth noting that glycans also play a structural and therefore in some sense or another presumably functional role with astrotactin-2.

5. The sMAC sample is commercially acquired. What was the sample composition, especially in terms of lipids, detergents and other buffer components? As this may materially impact on the data interpretation.

In all, this paper describes something of a detective story. The maps are carefully and thoughtfully dissected, and data iteratively analysed to define the density further in local detail. The paper is significantly strengthened in its inclusion of quantitative XL-MS data that greatly enhance interpretation due to evident signal for clusterin and vitronectin way above the background, enhancing the interpretations given, along with the level of detail recovered from analysis of locally sharpened maps. On which point, the quality of the maps strongly suggest that all approaches taken to density modification and analysis were appropriately selected.

Reviewer #2 (Remarks to the Author):

The manuscript by Menny et al. reports the structural analysis of the soluble precursor of the membrane attack complex (sMAC), an important complex for immune response, by cryo-EM and XL-MS.

The authors obtained high-resolution cryo-EM maps from complexes containing 1 to 3 copies of the C9 subunit, and confirmed subunit stoichiometries and positioning with the help of mass spectrometry and mass photometry. Based on these initial results, focused refinement of the cryo-EM data was used to look at specific features of the complex. First, interactions with the chaperones clusterin and vitronectin were studied in detail. Based on cross-linking restraints and lower resolution densities, clusterin was localized to the C9 subunits in the (C9)₃ complex. Second, the structure of the C7 subunit in the complex was refined further, revealing more details about the interface to the adjacent subunits. Finally, conformational transitions in C9 were studied in detail, allowing the formulation of a mechanistic hypothesis for pore formation by the sMAC complex.

Overall, this a very nice manuscript that is clearly written and provides very detailed information about the methods used to generate the underlying data. It represents a good example for the combination of high-resolution electron microscopy and chemical cross-linking, revealing their complementarity to better characterize dynamic and heterogeneous complexes as exemplified by the clusterin binding and the conformational transitions in C9. I have only few minor comments as summarized below.

Minor comments:

Page 24, line 538: Some of the search parameters are not really clear to me: Hexose on Trp? Why Tyr sulfation?

Page 24, line 540: 5% FDR at the PSM level is a quite lenient setting. Depending on the level of redundancy, the FDR at the peptide pair level may be considerably higher.

SI, Figure 4: Dashed grey lines on grey background are not really visible, I suggest to change the color for the incompatible cross-links.

It was quite challenging to find typos in the text, but I managed to find a few:

Line 537: Carbamicomethyl > Carbamidomethyl

SI, line 18: Informtation > Information

SI Figure 2, panel a, right: "cluterin" in the legend

SI, line 68: "AGC" does not seem to belong there

SI, line 115: glyan > glycan

SI, line 117: (c) should be (b)

SI, line 118: peforin > perforin

Reviewed by Alexander Leitner, ETH Zurich, Switzerland

Reviewer #3 (Remarks to the Author):

The manuscript describes cryoEM structures of soluble precursors of the membrane attack complex (sMAC). The regulations of the membrane attack complex must be very precise in order to prevent any bystander damage. Extracellular chaperones, such as clusterin and vitronectin, are important in this respect, since they associate with sMAC and can prevent its unwanted activity. This paper aims to reveal how this is achieved by using cryoEM and cross-linking linked with mass spectrometry. While analyzing the sMAC structure it also becomes apparent, that one of the C9 units has, interestingly, changed conformation of the part that is crucial for membrane perforation and this is discussed in a great detail and compared to other MACPF domain-containing proteins. Additional insights into the structure of MAC components is additionally provided (i.e. conformation of C7). Altogether data represent an improvement of our understanding of complement regulation by soluble chaperones and also provide deeper insight into how MACPF proteins form pores.

The paper is composed of two parts. One is dedicated to the analysis of the association of clusterin with the sMAC and the other one to the analysis of the C9 structure. I think the title should be changed to be more specific and better reflect what was done in the manuscript.

Supplementary Figure 3 reports the proteins in the sMAC according to the XL-MS analysis. In the panel, a, quite a large proportion of proteins represent other proteins, almost one quarter. Please explain what these proteins are and discuss their potential role. Since the interaction between clusterin and the sMAC is electrostatic in nature, is it likely that also some other proteins from the serum associate with the sMAC in a similar fashion?

Could the interaction of the sMAC and clusterin be verified by some biophysical approaches at different conditions (i.e. presence of various concentrations of salt), such as ITC or SPR? This could prove that the interaction is electrostatic in nature and provide quantitative data.

The dots in panels c and d of supplementary figure 3 are not visible well in all cases, please increase the size.

A lot of abbreviations are used unexplained in the legend to Supp. figure 4 (but also check for in other legends to figures in the manuscript, i.e. fig. 1), e.g. DMTMM, DSS, BN-PAGE, SDS-PAGE, etc.

Additional density that could be attributed to clusterin was identified by focusing on C9 part of the sMAC. CryoEM density, MS analysis and modeling of clusterin support position of one clusterin molecule at the top of C9 domains. MS data also shows that there are multiple copies of clusterin present on the sMAC. Some discussion should be devoted to the interaction of these other copies with the sMAC. It seems there are a lot of links of clusterin also with C7 and C8beta (Supp Fig 3d). Could focusing, with cryoEM analysis, on the C6, C7, C8beta region of the complex, instead of C9, reveal some additional unaccounted for by electron density that could be modeled to either clusterin or vitronectin?

Lines 185-186: it is claimed that clusterin engages with the sMAC through electrostatic interactions to block the propagation of polymerizing proteins. Why would propagation be blocked? Is it because clusterin obstructs surfaces involved in C9 oligomerization? A brief explanation based on cryoEM model (Fig. 2c) should be proposed.

Some of the proposals in the manuscript could be tested by mutagenesis. In particular, the structural transitions of TMH2 that are extensively characterized and presented in Fig. 4 and the role of P146 in disease. Is this feasible? This would supplement propositions about C9 oligomerization that is proposed on panel f of Fig. 4.

Structural basis for how sMAC is packaged for clearance
A. Menny & M.V. Lukassen et al.
Response to reviewers

We are writing in response to your e-mail message with peer-review comments on our manuscript "Structural basis for how sMAC is packaged for clearance" (NCOMMS-21-15430). We were pleased and encouraged by the reviewers' recognition of the interest and significance of our results. We thank the reviewers for their suggestions on how we could strengthen and improve the manuscript. We have now revised our manuscript to address the reviewers' concerns, as detailed below.

REVIEWER COMMENTS

Reviewer #1 (Remarks to the Author):

This is a good paper and it deserves to be published, after appropriate revision. The paper describes an important and insight-providing set of structures of the sMAC which significantly enhance our understanding of MAC biology and assembly.

A key starting point is to recognise that this paper concerns "an immune activation complex that is formed from MAC assembly precursors released into plasma and scavenged by blood-based chaperones". Thus we are dealing both with a complex which exists in serum and which is of relevance to the initiation of complement membrane attack, and with a complex which at the same time reveals details of relevance to piecemeal assembly of the MAC on a target membrane, one subunit at a time. That in any case is what I took from the paper as a whole, though I felt the emphasis was initially on the first aspect - membrane attack by sMAC/its prevention by vitronectin and clusterin - and only latterly on the insights provided for on-the-membrane complex assembly. Clearly, the particular focus on this first aspect may relate to the clinical relevance of the study - e.g. to ARMD - but I guess as with any structural analysis whatever the results there is likely to be something of relevance to more than one aspect of the system's functioning. In this case, most strikingly, to the structural stability of MAC complexes (sMACs) with their hydrophobic beta hairpins deployed but which nevertheless are (i) free in solution but (ii) can partition into a target bilayer (because why else the need for clusterin and vitronectin?).

Some specific comments:

1. The densities for clusterin and vitronectin prove hard to define. I don't see a worrisome issue here - see below - but (page 5, end of para. 1) I wondered if some 2d classification of negatively stained images could give a bit of an insight into the different states adopted by the bound clusterin and vitronectin subunits? This may indicate if there are a relatively small number of statically disordered states or a larger and potentially undefinable set of dynamic conformations. If static then the different states may have some functional relevance.

Response: Our mass photometry data show that sMAC is a heterogeneous complex that varies not only in the number of copies of C9, but also exhibits significant heterogeneity in the number of clusterin and vitronectin molecules that can comprise the complex (Fig. 1d). It is important to note that the size of these chaperones in their monomeric states are less than a 1/10th of the molecular mass of the complex. It was only by separating particles based on the stoichiometry of C9 that we were able to dig deeper into the complexity of sMAC heterogeneity and detect density for chaperones. Classification of negatively-stained particles can be influenced by staining artefacts and granularity of the stain. These issues are particularly relevant when interpreting small differences in protein structures; therefore, we respectfully disagree with the use of this approach. We have already performed a global

3D classification of the complex where any large statically ordered states would have been picked up.

2. Be clear in the figure legend 2d that the clusterin depiction is based on a homology model as described on page 8 and in the Methods, and not on an experimental structure.

Response: This is now included in the legend of Figure 2b. The legend for 2d now explicitly references back to the model shown in 2b.

“Structural homology model for the clusterin core with intra-molecular clusterin cross-links derived from XL-MS mapped (black lines)”.

“Coulombic electrostatic potential ranging from -10 (red) to 10 (blue) kcal/(mol·e) calculated from the models for complement proteins in 3C9-sMAC (bottom surface) and the model for clusterin shown in 2b (top surface)”

3. A critical thing to address is the unfurling of the beta hairpins in solution. Generally they are considered in MACPF/CDC proteins to unfurl on membranes - here in the structure determined they are already unfurled (though only partly in the case of the second copy of C9). This suggests that the MAC can insert in an already partly pore-forming state (an arc of C5b-9 up to C5b-999) and it then extends in size in the membrane with the recruitment of additional copies of C9. This is an important observation and entirely feasible. A good comparator would be the likely spontaneous insertion (at least according to the assisted model) of small (I would argue) beta sheeted porins by the BAM complex. [In my view of the two models for BAM the templating model might likely apply to larger barrels, the assisted model to smaller ones.] Another would be the work of Mark Wallace and colleagues on alpha-hemolysin which suggests ways in which membranes remodel themselves when confronted with inserting beta barrel structures.

It seems to me that the paper discusses the insights more in terms of what would happen with piecemeal assembly of the MAC on the target membrane, but the actual structure determined relates to a different scenario. It casts light on both processes - step by step and insertion at once of C5b-9n where n=1-3 - but the relevance to both should I think be teased out more, with comment on the striking stability of the unfurled TMHs for C6-C9 included.

Response: We thank the reviewer for the opportunity to expand our discussion of how sMAC could inform a broader mechanistic insight into beta-barrel pore formation. We now include a paragraph that discusses our structure within the context of initiation and propagation of beta-barrel pores referencing BAM and a recent study by Mark Wallace.

“In our structure of sMAC, TMH residues of complement proteins are stably unfurled into their β -hairpin conformation even in the absence of a lipid bilayer. Therefore, it may be possible that an arc comprised of C6, C7, C8 and up to 3 copies of C9 could serve as a minimal nucleation unit that inserts into the membrane to initiate MAC pore formation. Indeed, bacterial homologues of the cholesterol-dependent cytolysin superfamily can form oligomeric arcs that remodel lipid bilayers, generating membrane-inserted assembly intermediates that are non-conductive³⁸. To complete the MAC pore, further C9 molecules could then be sequentially inserted by templating leading edge β -strands, reminiscent to the folding of β -sheet porins by the b-barrel assembly machine (BAM)³⁹.”

4. On page 12 the role of glycans is discussed and the relationship between the glycans found in perforin-2/MPEG1 and C9 is described as "analogous". Is not the relationship specifically "homologous"? In general, the proposal of importance for the glycans in MAC

structure/mechanism is insightful and welcome - on this point it may be worth noting that glycans also play a structural and therefore in some sense or another presumably functional role with astrotactin-2.

Response: the relationship between C9 and MPEG1 is now explicitly described as homologous:

“We note that within the pore conformation, MPEG-1 contains a glycan on the leading edge of TMH237, homologous to C9.”

We also now mention the role of glycans for perforin-1 and astrotactin-2, other MACPF-containing proteins.

“More broadly, glycans of MACPF-containing proteins are shown to have a structural role in astrotactin-2⁴⁰ and are functional regulators of perforin-1 activity⁴¹.”

5. The sMAC sample is commercially acquired. What was the sample composition, especially in terms of lipids, detergents and other buffer components? As this may materially impact on the data interpretation.

Response: sMAC, provided by the commercial supplier, was activated in the absence of lipid bilayer membranes as stated in their product description. The buffer it is supplied in contains no lipids or detergents: 10 mM sodium phosphate, 145 mM NaCl, pH 7.3. This information is now provided in the materials and methods section.

“To prepare cryoEM grids, sMAC (Complement Technologies) provided in 10 mM sodium phosphate, 145 mM NaCl, pH 7.3 was diluted to 0.065 mg/ml in 120 mM NaCl, 10 mM Hepes pH7.4 and used for freezing within the hour.”

In all, this paper describes something of a detective story. The maps are carefully and thoughtfully dissected, and data iteratively analysed to define the density further in local detail. The paper is significantly strengthened in its inclusion of quantitative XL-MS data that greatly enhance interpretation due to evident signal for clusterin and vitronectin way above the background, enhancing the interpretations given, along with the level of detail recovered from analysis of locally sharpened maps. On which point, the quality of the maps strongly suggest that all approaches taken to density modification and analysis were appropriately selected.

Response: We thank the reviewer for this positive overall assessment of our manuscript.

Reviewer #2 (Remarks to the Author):

The manuscript by Menny et al. reports the structural analysis of the soluble precursor of the membrane attack complex (sMAC), an important complex for immune response, by cryo-EM and XL-MS.

The authors obtained high-resolution cryo-EM maps from complexes containing 1 to 3 copies of the C9 subunit, and confirmed subunit stoichiometries and positioning with the help of mass spectrometry and mass photometry. Based on these initial results, focused refinement of the cryo-EM data was used to look at specific features of the complex. First, interactions with the chaperones clusterin and vitronectin were studied in detail. Based on cross-linking restraints and lower resolution densities, clusterin was localized to the C9 subunits in the (C9)₃ complex. Second, the structure of the C7 subunit in the complex was refined further, revealing more details about the interface to the adjacent subunits. Finally, conformational transitions in C9 were studied in detail, allowing the formulation of a

mechanistic hypothesis for pore formation by the sMAC complex.

Overall, this a very nice manuscript that is clearly written and provides very detailed information about the methods used to generate the underlying data. It represents a good example for the combination of high-resolution electron microscopy and chemical cross-linking, revealing their complementarity to better characterize dynamic and heterogeneous complexes as exemplified by the clusterin binding and the conformational transitions in C9. I have only few minor comments as summarized below.

Response: We thank the reviewer for this positive overall assessment of our manuscript.

Minor comments:

Page 24, line 538: Some of the search parameters are not really clear to me: Hexose on Trp? Why Tyr sulfation?

Response: These modifications were added to the search to optimize the identification of cross-linked peptides within the sMAC complex. The TSP1 domains of C6-C9 are known to have several C-mannosylated Trp residues (hexose on Trp) and vitronectin have several Tyr sulfation sites. We added a sentence to the methods section to clarify this:

“Carbamidomethyl (C) was set as fixed modification and oxidation (M), and acetyl (protein N-term) as variable modifications. Furthermore, we added hex (W) and sulfo (Y) as variable modifications to account for c-mannosylation of thrombospondin type 1 domains and sulfation of vitronectin.”

Page 24, line 540: 5% FDR at the PSM level is a quite lenient setting. Depending on the level of redundancy, the FDR at the peptide pair level may be considerably higher.

Response: A previous study showed that the performance of pLink at 1% and 5% FDR are highly similar (Zhao et al., 2020). This is also the case for our data. As an example, we searched the DMTMM cross-linked data with both 1% and 5% FDR. Here, only 11 of 202 cross-links (at 5% FDR) were not identified at 1% FDR. Our filtering of cross-links included in the analysis is also very stringent since we only include cross-links identified in all triplicates. Thus, we are confident in the cross-links reported in the study.

Zhao et al., Use of multiple ion fragmentation methods to identify protein cross-links and facilitate comparison of data interpretation algorithms. *J. Proteome Res.* 19:7, 2758-2771 (2020).

SI, Figure 4: Dashed grey lines on grey background are not really visible, I suggest to change the color for the incompatible cross-links.

Response: The colors of the incompatible cross-links were changed to pink to improve visibility.

It was quite challenging to find typos in the text, but I managed to find a few:

Line 537: Carbamicomethyl > Carbamidomethyl

SI, line 18: Informtation > Information

SI Figure 2, panel a, right: "cluterin" in the legend

SI, line 68: "AGC" does not seem to belong there

SI, line 115: glyan > glycan

SI, line 117: (c) should be (b)

SI, line 118: peforin > perforin

Reviewed by Alexander Leitner, ETH Zurich, Switzerland

Response: We thank the reviewer for their careful reading of our manuscript. These typos have now been changed.

Reviewer #3 (Remarks to the Author):

The manuscript describes cryoEM structures of soluble precursors of the membrane attack complex (sMAC). The regulations of the membrane attack complex must be very precise in order to prevent any bystander damage. Extracellular chaperones, such as clusterin and vitronectin, are important in this respect, since they associate with sMAC and can prevent its unwanted activity. This paper aims to reveal how this is achieved by using cryoEM and cross-linking linked with mass spectrometry. While analyzing the sMAC structure it also becomes apparent, that one of the C9 units has, interestingly, changed conformation of the part that is crucial for membrane perforation and this is discussed in a great detail and compared to other MACPF domain-containing proteins. Additional insights into the structure of MAC components is additionally provided (i.e. conformation of C7). Altogether data represent an improvement of our understanding of complement regulation by soluble chaperones and also provide deeper insight into how MACPF proteins form pores.

The paper is composed of two parts. One is dedicated to the analysis of the association of clusterin with the sMAC and the other one to the analysis of the C9 structure. I think the title should be changed to be more specific and better reflect what was done in the manuscript.

Response: We thank the reviewer for this positive overall assessment of our manuscript. As the analysis of the C9 structure is directly related to how clusterin traps the complex and prevents pore formation, we would respectfully disagree with the reviewer's request to change the title of the manuscript.

Supplementary Figure 3 reports the proteins in the sMAC according to the XL-MS analysis. In the panel, a, quite a large proportion of proteins represent other proteins, almost one quarter. Please explain what these proteins are and discuss their potential role. Since the interaction between clusterin and the sMAC is electrostatic in nature, is it likely that also some other proteins from the serum associate with the sMAC in a similar fashion?

Response: Panel a of Supplementary Fig. 3 is a bottom-up MS analysis of the sMAC sample and is not related to the XL-MS analysis. The sMAC components are the most abundant proteins in the sample and the other 23% is representing 112 different proteins. The sMAC complex is purified from plasma and it is expected that there will be some contaminants present in trace amounts. Due to high sensitivity of MS these trace amounts will be detected. The only other proteins showing an indication of interaction to the complement components in our XL data are FAT1 and thrombin (Supplementary Figure 3b). These cross-links were only found in the DSS data and was of low abundance. We do not find speculations regarding these potential interactions relevant for our study.

Could the interaction of the sMAC and clusterin be verified by some biophysical approaches at different conditions (i.e. presence of various concentrations of salt), such as ITC or SPR?

This could prove that the interaction is electrostatic in nature and provide quantitative data.

Response: It is well documented that clusterin binds complement proteins only when the terminal pathway is activated and prevents further oligomerization of C9 (Tschopp et al., 1993). There is no evidence that clusterin can be added exogenously to preformed and isolated MAC precursors. Furthermore, polymerization of MACPF-containing complement proteins is also driven by electrostatic complementarity (Menny et al., 2018). Any experiments performed at high salt concentrations may impact the integrity of the complement complex as well as the interaction with clusterin. Based on these data, we conclude that an *in vitro* biophysical characterization investigating electrostatic interactions of sMAC would be extremely challenging to interpret and is beyond the scope of this study.

Tschopp J, Chonn A, Hertig S, French LE. Clusterin, the human apolipoprotein and complement inhibitor, binds to complement C7, C8 beta, and the b domain of C9. *J Immunol* 151, 2159-65 (1993).

Menny A, Serna M, Boyd CM, Gardner S, Joseph AP, Morgan BP, Topf M, Brooks NJ, Bubeck D. CryoEM reveals how the complement membrane attack complex ruptures lipid bilayers. *Nature Commun.* 9, 5316 (2018).

The dots in panels c and d of supplementary figure 3 are not visible well in all cases, please increase the size.

Response: We have increased the sizes of spheres representing interacting residues for better visibility.

A lot of abbreviations are used unexplained in the legend to Supp. figure 4 (but also check for in other legends to figures in the manuscript, i.e. fig. 1), e.g. DMTMM, DSS, BN-PAGE, SDS-PAGE, etc.

Response: These abbreviations are now defined in the legends to Figure 1, Supplementary Figure 4, and the methods section.

Additional density that could be attributed to clusterin was identified by focusing on C9 part of the sMAC. CryoEM density, MS analysis and modeling of clusterin support position of one clusterin molecule at the top of C9 domains. MS data also shows that there are multiple copies of clusterin present on the sMAC. Some discussion should be devoted to the interaction of these other copies with the sMAC. It seems there are a lot of links of clusterin also with C7 and C8beta (Supp Fig 3d). Could focusing, with cryoEM analysis, on the C6, C7, C8beta region of the complex, instead of C9, reveal some additional unaccounted for by electron density that could be modeled to either clusterin or vitronectin?

Response: We thank the reviewer for allowing us the opportunity to further explore other chaperone binding sites near regions of the map corresponding to C6, C7 and C8. We have now performed multibody analysis on our 3C9 sMAC map. In a multibody analysis, each defined body is used to perform density subtraction on the raw particles followed by a 3D refinement of each body (Nakane et al., eLife 2018). Specifically, we have defined 2 bodies: 1 corresponding to the terminal two C9s and the second (shown below) corresponding to C5b (light grey), C6 (blue), C7 (orange), C8 (alpha, beta and gamma in shades of purple) and one C9 (green). In doing so, we resolve extra density near the observed crosslinks of C8 and vitronectin (dark grey density, circled in red), and density above the LDL domains of C7 and C8 (dark grey density, black circle), similar to the clusterin position above C9. Other areas of difference between the map and model for complement proteins (dark grey) near C6 (blue) are likely attributed to known glycosylation sites not included in our PDB model. Given that these extra density features from the multibody analysis are weak and still too low

resolution to model either clusterin or vitronectin, we would argue that the structural data does not add additional information above the XL-MS data and may detract from the more conclusive structural data already presented. Therefore, we have decided not to include this new analysis in the manuscript.

Nakane T, Kimanius D, Lindahl E, Scheres SHW. Characterisation of molecular motions in cryo-EM single-particle data by multi-body refinement in RELION. *eLife*. 7:e36861 (2018).

It is important to note that our homology model for clusterin only includes the core of the molecule, with long flexible extensions left unmodeled. It may be possible that these residues extend beyond the core to interact with C6, C7, and C8. We have now explicitly discussed the possible interactions of these unmodeled residues and more distal complement proteins.

“Beyond this core, we find that clusterin makes additional cross-links with a range of complement proteins distal from C9 (Supplementary Fig. 3). Indeed, the ensemble of models produced by trRosetta reveals long extended domains that flexibly hinge from the core (Supplementary Fig. 6). It is possible that these extensions could interact with distal complement proteins.”

We have already provided a possible role for additional clusterin binding sites in the discussion.

“While the role of this second binding site in regulating pore formation remains unclear, it may be important in linking cargo with an endocytic receptor for clearance.”

Lines 185-186: it is claimed that clusterin engages with the sMAC through electrostatic interactions to block the propagation of polymerizing proteins. Why would propagation be blocked? Is it because clusterin obstructs surfaces involved in C9 oligomerization? A brief explanation based on cryoEM model (Fig. 2c) should be proposed.

Response: We have now fitted an oligomer of four C9 molecules from the deposited MAC structure (PDB ID: 6H04) into the 3C9 focused refined map. In doing so, we observe a steric clash between density attributed to clusterin and the fourth C9 molecule in the oligomer. We included this analysis in a supplementary figure (Supplementary Fig. 5) to clarify how density for clusterin obstructs a surface involved in C9 oligomerization.

“This density caps a similarly negatively charged polymerizing face of the C9 MACPF (Fig. 2c and d) and sterically obstructs additional C9 binding (Supplementary Fig. 5).”

Some of the proposals in the manuscript could be tested by mutagenesis. In particular, the structural transitions of TMH2 that are extensively characterized and presented in Fig. 4 and the role of P146 in disease. Is this feasible? This would supplement propositions about C9 oligomerization that is proposed on panel f of Fig. 4.

Response: Since submitting this manuscript, we have published (June 2021) mutational data characterizing the role of P146 in C9 polymerization and now include a citation for this work which supports our structural model.

“Our structural model is further supported by mutational studies showing that mutation of Pro₁₄₆ increases C9 polymerization³⁵.”

McMahon O, Hallam TM, Patel S, Harris CL, Menny A, Zelek WM, Widjajahakim R, Java A, Cox TE, Tzoumas N, D H W Steel, Shuttleworth VG, Smith-Jackson K, Brocklebank V, Griffiths H, Cree AJ, Atkinson JP, Lottery AJ, Bubeck D, Morgan BP, Marchbank KJ, Seddon JM, Kavanagh D. The rare C9 P167S risk variant for age-related macular degeneration increases polymerization of the terminal component of the complement cascade. *Human Molecular Genetics*. 30(13):1188-1199 (2021).

REVIEWERS' COMMENTS

Reviewer #1 (Remarks to the Author):

I am grateful to the authors for their thoughtful and careful responses to the points I raised in my previous assessment. I am glad to recommend publication of the paper as it is.

Reviewer #2 (Remarks to the Author):

The authors addressed all my comments in this revised version of the manuscript. I have no further requests.

Reviewed by Alexander Leitner, ETH Zurich

Reviewer #3 (Remarks to the Author):

The authors have replied satisfactorily to comments, I do not have any further issues.